# Thoracic UltrasONOgraphy Reporting: The TUONO Study

**DOI:** 10.3390/jcm11237126

**Published:** 2022-11-30

**Authors:** Italo Calamai, Massimiliano Greco, Stefano Finazzi, Marzia Savi, Gaia Vitiello, Elena Garbero, Rosario Spina, Andrea Montisci, Silvia Mongodi, Guido Bertolini

**Affiliations:** 1Anesthesia and Intensive Care Unit AUsl Toscana Centro, Ospedale San Giuseppe, Viale Boccaccio 16/20, 50053 Empoli, Italy; 2Department of Biomedical Sciences, Humanitas University, 20090 Milan, Italy; 3Department of Anesthesiology and Intensive Care, IRCCS Humanitas Research Hospital, 20100 Milan, Italy; 4Laboratory of Clinical Data Science, Mario Negri Institute of Pharmacological Research IRCCS, 24020 Ranica, Italy; 5Division of Cardiothoracic Intensive Care, Cardiothoracic Department, ASST Spedali Civili, 25121 Brescia, Italy; 6Anaesthesia and Intensive Care, San Matteo Hospital, 27100 Pavia, Italy; 7Laboratory of Clinical Epidemiology, Mario Negri Institute of Pharmacological Research IRCCS, 24020 Ranica, Italy

**Keywords:** lung, ultrasound, critical care, point-of-care, surveys and questionnaires

## Abstract

Lung ultrasound (LUS) is a validated technique for the prompt diagnosis and bedside monitoring of critically ill patients due to its availability, safety profile, and cost-effectiveness. The aim of this work is to detect similarities and differences among LUS reports performed in ICUs and to provide a common ground for an integrated report form. We collected all LUS reports during an index week in 21 ICUs from the GiViTI network. First, we considered signs, chest areas, and terminology reported. Then, we compared different report structures and categorized them as structured reports (SRs), provided with a predefined model form, and free unstructured text reports (FTRs) that had no predetermined structure. We analyzed 171 reports from 21 ICUs, and 59 reports from 5 ICUs were structured. All the reports presented a qualitative description that mainly focused on the presence of B-lines, consolidations, and pleural effusion. Zones were defined in 66 reports (39%). In SRs, a complete examination of all the regions was more frequently achieved (96% vs. 74%), and a higher impact on therapeutic strategies was observed (17% vs. 6%). LUS reports vary significantly among different centers. Adopting an integrated SR seems to promote a systematic approach in scanning and reporting, with a potential impact on LUS clinical applications.

## 1. Introduction

Lung ultrasound (LUS) has emerged as a highly sensitive and specific tool for the diagnosis of acute lung diseases both in Intensive Care Units (ICUs) and in ordinary wards.

In the last decade, its reliability for the initial diagnosis of acute respiratory failure has been displayed [1], and it has become a valuable monitoring tool in the critically ill [2]. However, the lack of a standardized reporting method may limit reproducibility and transmission of findings within intensive care teams. LUS findings are not always written or reported in a structured way with a complete and reproducible description of the investigated regions, potentially limiting the clinical utility of LUS examination [3].

The semeiology of LUS has been exhaustively described [4]. Although its nomenclature, techniques, and indications have been established since 2012 [5], LUS reporting has not been appropriately standardized until now, and real-world approaches to LUS reporting in ICUs differ from center to center.

This study aims to investigate the characteristics of LUS reporting in clinical practice in a large Italian ICU network, focusing on a comparison between structured and unstructured reports to identify potential benefits.

## 2. Materials and Methods

We conducted a multicenter observational study in 2018–2019 involving 21 Italian ICUs from the GiViTI ICU network [6]. The network was composed of centers where LUS is regularly performed in clinical practice and included both academic-tertiary center hospitals and non-academic centers.

GiViTI is the largest Italian ICUs network whose primary purpose is to improve the quality of care by sharing the analysis of data from more than 190 participating centers.

We collected all the LUS reports performed in the enrolled ICUs over 7 days (“index week”) selected by each center in the timespan between 25 January 2018 and 17 November 2019. Data were collected and reported by the center on a predefined form in a dedicated section of Prosafe software (Prosafe Core, GiVITI, Ranica, Bergamo, Italy), which is a software created by GiVITI for data collection for quality improvement and scientific research in intensive care. For this study, a new Prosafe software interface stem was built to allow centralized reports collection while preserving the exact structure of the reports.

During the index week, all the participating centers were required to send all the LUS exams performed and documented in clinical record, even if LUS reporting was partial or incomplete.

We analyzed the type of the report and identified reports as structured reports (SRs) if they had a predefined structured that required systematic filling or as free-text reports (FTRs) if they had no structure and were eventually integrated in the patient’s daily report. We noted qualitative and quantitative description of the findings, the presence of other associated ultrasound investigations, including the reporting of ICU admission diagnosis, the clinical purpose for the LUS examination, and the type of probe. We considered reports as *investigated* if all the structures, features, and signs were mentioned or ruled out, while *non-investigated* reports did not explicitly mention these areas.

Localization of the findings was distinguished into predefined zones (2, 3, 4, or 6 per side), undefined zones (right or left side localization, anatomical reference, and mainly or only lung bases descriptions), and vague description (e.g., regarding all the lungs or the remaining lungs).

Associated ultrasound examinations, such as diaphragm characteristics and echocardiography, were reported, if performed. The impacts on clinical management in terms of the definition of the diagnostic and therapeutic path was also noted.

FTRs were compared to SRs by filling a predefined form with n° 91 items (Appendix A).

The study obtained institutional review board approval in all participating centers.

We reported ordinal and continuous variables as mean and standard deviation (SD) or median and interquartile range (IQR) as adequate, while numbers (percentages) were used for categorical variables. The Fisher Exact test was used to identify statistically significant differences between SRs and FTRs.

## 3. Results

### 3.1. Overview

We analyzed 171 LUS reports from 21 ICUs. The number of LUS examinations performed by each ICU during the index week, expressed as a mean and an interquartile range, was 8 [1,2,3,4,5,6,7,8,9,10,11,12,13,14,15,16,17,18,19]. The main characteristics of the enrolled patients are shown in Table 1.

Fifty-nine reports out of 171 were SRs from 5 ICUs (with a range of 5–19 per ICU), while 112 were FTR from 16 ICUs (with a range 1–13). All five ICUs who adopted a SR provided a free-text area for writing comments. Among them, four adopted a template with predefined pulmonary zones as fixed items and the possibility to describe the corresponding LUS findings through free or prearranged text. In contrast, one ICU used a template reporting predefined LUS signs as fixed items with their localization to be described.

### 3.2. Purpose of LUS and Findings Reported

Most reports aimed to describe all the findings encountered by LUS investigation.

Only 21 (11%) reports were performed to *rule out* specific conditions (SRs = 2 [3%] vs. FTRs = 19 [15%], *p* = 0.021). Pleural effusion was the most frequently ruled out condition (Appendix A).

The pleura was investigated overall in 102 reports (60%), and pleural sliding was described in 92 (90%) of these reports. The pleural characteristics were described in 28 reports (27%).

Parenchymal features relating to one or more zones examined in a single report are represented in Figure 1 and Table 2.

A total of 96 (56%) reports described B lines. The number of B lines were reported in 32% of the reports, and their aspect (separated or coalescent) was reported in 17% of the reports. A semiquantitative description (few, many, some, etc.) was present in 39% of the reports. The Lung Ultrasound score (LUS score) was used in only 4 FTR (2.3%) for the grading of lung aeration and water content. Pulmonary water content was estimated in 12% of the reports.

The presence of effusion was described in 69 reports (40%), and in almost all of them (96%), its characteristics were further specified. Pleural effusions were qualitatively described as small, large, or abundant in 64% of the cases. Effusion was measured in centimeters in 45% of the reports, and its volume was estimated in 20% of them. Description of appearance and/or echogenicity (e.g., anechoic, hypoechoic, echogenic, or septated effusion) was rare (7%), as reported in Table 2.

A tissue-like pattern was present in 82 reports (48%). It was identified as a consolidation in 56 reports, while in 47 (57%) reports, specific conditions were reported, including atelectasis (23–28%), pneumonia (1–1.2%), or atelectasis and/or pneumonia (23–28%).

Consolidations (*n* = 56) were qualitatively described as small or large in 14% of the cases. The examiners reported the measurement of consolidation in cm in only 4% of the cases. Atelectasis was defined as minimal, initial, small, large, extended, etc. in 23% of the cases. Dimensions of pneumonia (total *n* = 12) were never estimated.

The presence of bronchogram was reported in 20/82 cases (24%) of consolidations, distinguishing static from dynamic bronchogram in about 50% of the cases.

Subpleural consolidations were reported in 13 reports (8%) and were generally associated with B lines (12 reports—92%) or lung consolidations (7 reports—54%).

Localization of findings varied from vague (i.e., on the left or on the right hemithorax) to definite (specified lung areas or crossing of anatomical lines). Overall, a total of 66 reports (39%) defined the examined lung zones.

### 3.3. Comparing SRs to FTRs

Some findings were more commonly reported in SR than in FTR, as shown in Table 2.

Reporting B lines and pleural effusions without further specification was more frequent in SRs (*p* < 0.0109; *p* = 0.0294), while a semiquantitative description of B lines and effusions was more frequent in FTRs (*p* = 0.0555; *p* = 0.0363). Estimation of pleural effusion volume was more frequent in SRs (*p* < 0.0001), as well as in ICU admission diagnosis (58% vs. 12%), reason for LUS examination (68% vs. 4%), type of probe (68% vs. 12%), and type of mechanical ventilation (61% vs. 31%), with *p* < 0.001 for all comparisons.

The rate of areas described as normal was significantly higher in SRs than in FTRs (*p* < 0.0001).

Zones were more frequently defined in SRs (80%) than FTRs (17%) (*p* < 0.0001). A partial investigation (at least one missing area) or a vague localization of findings were less frequent in SRs (5%) than FTRs (25%) (*p* < 0.001).

### 3.4. Clinical Considerations

LUS findings were compared with previous LUS examinations in 28 reports (16%), and their modifications according to changes in ventilation were described in 11 reports (6%).

Clinical conclusions (63 vs. 6%, *p* = <0.0001) were reported ten times more in SRs than in FTRs. SRs outlined the need for further imaging investigations more frequently (63 vs. 19%, *p* = 0.0002).

LUS results guided the implementation of new therapeutic strategies, mostly involving ventilation, CRRT, antimicrobial, and diuretic therapy, in 19% of SRs and 6% of FTRs (*p* = 0.01).

## 4. Discussion

In this study, we described how differently LUS findings were reported among a network of general ICUs in Italy for the first time. A minority of ultrasound reports were recorded according to a systematic approach, while most reports analyzed were free text. Reports mainly focused on the qualitative description of the presence/absence of ultrasound findings. We pointed out a large variability in quantitative and qualitative findings reported in real practice. The adoption of a SR seems to offer some advantages in terms of completeness of reporting, systematic approach to zones exploration, and integration with clinical findings.

The value of LUS in emergency and ordinary activities has been known for years and is definitively sanctioned by its use during the ongoing COVID-19 pandemic [7]. Many aspects of LUS have been defined by Volpicelli et al. [5], but recommendations regarding how to report LUS exams have not been established yet. Lung ultrasound reporting is still an open field for research. To our knowledge, only few papers dealt with this item [8,9]. Both the models proposed different but standardized strategies to pioneer a systematic reporting of LUS. Tutino adopted an operative checklist, while Via proposed a visual intuitive report. Tutino et al. observed that the introduction of a standardized electronic recording sheet improved the uniformity of reporting and the completeness and accuracy of ultrasound examinations among different operators, favoring the evaluation of all the parameters required for a complete exam and report. The main limitation of that study was the lack of a control group to effectively compare standardized vs. non standardized reports and correlate the association between electronic sheet introduction and LUS quality improvement.

In our study, FTRs presented vague localization or unmentioned lung areas investigation more frequently, suggesting the potential role of SRs in addressing more thorough examination and more accurate reporting.

Via et al. presented a simplified SR format with six areas per side to be investigated. They defined pleural and parenchymal findings to be sought and attributed a number-coded rating of findings to calculate the LUS score. They adopted a SR to allow a much more rapid reporting of diagnostic, screening, monitoring, and procedure-guiding examinations and to favor the evaluation of diseases over time by repeated systematic LUS. They endorsed implementing LUS reporting using a standardized approach based on common language and uniform staff training.

In the last decade, other LUS models with variable numbers of zones have been adopted to investigate interstitial syndrome or proposed to investigate and rule out other lung diseases [10,11,12,13]. Most of the ICUs participating in our study adopted the 6-zone model.

Few signs were commonly used to describe lungs and pleural findings, including lung sliding, A lines, B lines, consolidations, and effusions. Other terms proposed by the literature (*lung pulse*, *shred sign*, *bat sign*, *seashore sign*, *sinusoid signs*, *quad sign*, *curtain sign*, *and stratosphere sign*) were never or very rarely mentioned in the reports.

Lung sliding was the most investigated pleural sign. Other features, such as the regularity/irregularity of the pleural line, which helps differentiate inflammatory conditions, such as ARDS or COVID-19 pneumonia, from a non-inflammatory condition (AHF), were rarely reported.

We hypothesize that at the time of our study, the pleural aspect was much more neglected when compared to its current relevance due to the COVID-19 pandemic [14], during which it has been extensively investigated and associated with newly introduced signs (*light beam*) for an early diagnosis [15]. Additionally, the LUS score, which is a specific tool to quantify the loss of aeration, was used only in selected cases among the examined reports, but it has undergone widespread use to monitor COVID-19 pneumonia progression [16].

B lines, which are vertical artifacts increasing in number alongside the decrease in pulmonary air content (pneumonia and pulmonary edema) regularly underwent a semi-quantitative (e.g., few or many) and qualitative (e.g., separate or crowding) analysis. Conversely, consolidation dimensions (e.g., small, large, and initial) were occasionally described and rarely measured. Differentiation between static bronchogram (associated with atelectasis) and dynamic bronchogram (associated with pneumonia) were seldom performed. These findings may reflect a steeper learning curve for the quantification of consolidation than for B-line, [2] or a lower reproducibility of consolidation measure in clinical practice. Accordingly, atelectasis is predominantly distinguished from consolidation based on clinical evaluation rather than on the behavior of bronchogram.

Effusions were generally described in a semi-quantitative way (e.g., *small*, *large*, *abundant*, or *minimal*), while a clear estimation of effusion volume was less frequent. Quantification of effusion volume is pivotal in the decision-making of pleural drainage [17,18,19]. These results should alert intensivists about the need for recognized keystones in LUS reporting as quantification of findings may be useful for immediate decision making and monitoring the evolution of the effusion over time. From this perspective, SR may be the magic bullet toward the improvement of performing and reporting. Several studies have highlighted the superiority of SRs in completeness, accuracy, and ease of data sharing compared to FTRs [20,21]. Filling in pre-established sections and checklists can reduce the probability of underreporting essential elements.

However, a poorly conceived SR could lead to systematic errors in reporting. In our sample, B lines, consolidations, and effusions presence/absence without a quantitative definition were described more frequently in SRs than in FTRs. In this case, the template design probably limited the description of the findings and lowered the report quality.

The LUS report should be conceived for clinical data sharing, and it should be complete, clear, reproducible, and designed to meet clinicians’ demands. The latter aspect is fundamental as the indication for LUS is often not a complete examination but the need to rule out an emergent condition, and the fact that complete examination is not always feasible or useful due to time constraints cannot be neglected [22]. In contrast, during routine examination, LUS should be as complete as possible, balancing the complexity of collecting data with the ease of writing and reading. Some examples derived from our results are the following: a report should describe the pleura and lungs, either normal or pathological, while including practical and easy methods to quantify lung findings, such as a definition of B-lines as *separate or bundled*.

For the effusions, terms such as *mild, moderate,* and *abundant* should correspond to a predefined volume, estimated by various techniques. In adults, measuring a consolidation’s axis may be challenging, but it is fundamental to describe whether an air bronchogram is absent or present and its characteristics (static or dynamic). Our study shows room for improvement in LUS practice, specifically in the quantification of findings, and it suggests prompt evaluation of further strategies to improve the measurement of findings, including specific training regarding measurement and the inclusion of quantification in standardized reports.

In our findings, LUS reports were conducted at the same time as cardiac (17%) or diaphragmatic (19%) POCUSs. Associating LUS to diaphragmatic and cardiac POCUS may be an opportunity to optimize the evaluation and treatment of critically ill patients, approaching the interaction between heart and lungs from an ultrasonography perspective.

This study has some limitations. The lack of a threshold for the number of reports per ICU may have overrepresented the units with a higher LUS performance rate, which may have influenced the results, especially regarding the prevalence of one type of report over the other. Additionally, different operators may have produced different types of reports within the same ICU.

The number of ICUs is limited which is a factor that may under-represent the reality of ICU LUS practice.

Despite these limitations, this is, to our knowledge, the first study on LUS reports in a real-world ICU setting where lung ultrasound is routinely used in daily care, as demonstrated by the number of examinations collected in a week.

## 5. Conclusions

We compared the type of LUS reports (structured vs. unstructured) and highlighted how the structure of SRs might directly influence the clinical evaluation of patients. From the perspective of inserting LUS in the panel of daily examination tools, we suggest that LUS reports should accurately describe each pulmonary zone, even in the absence of pathological findings, to allow comparisons with previous and subsequent exams.

In this context, the widespread adoption of SR for LUS could play a pivotal role in increasing the clinical utility of LUS examination and in monitoring the evolution of lung conditions over time.

## Figures and Tables

**Figure 1 jcm-11-07126-f001:**
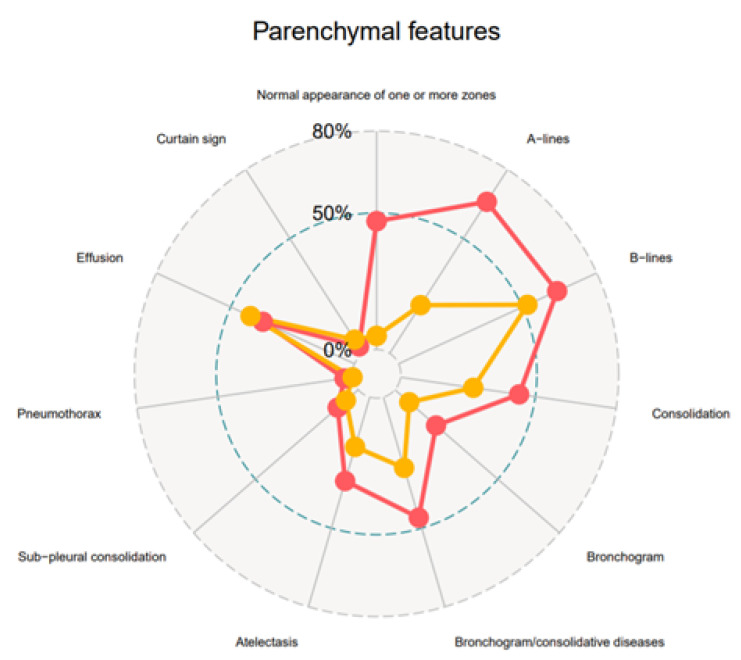
Spider web chart representing the main parenchymal features reported. Values are expressed as *n* (%).

**Table 1 jcm-11-07126-t001:** Characteristics of the population.

Characteristics	*n* (%)
Patients	106
Sex—male	69 (65%)
Age (y, SD)	68 ± 15
SAPS II score	34 ± 16
SOFA score	4 ± 3.6
ICU length of stay (d, SD)	7.1 ± 10.4
Mortality during ICU length of stay	9 (8.7%)
Admission unit:	
- Medical ward	53 (56%)
- Surgical ward	53 (56%)
Purpose of admission:	
- Monitoring/weaning	42 (39%)
- Intensive treatment	64 (64.5%)
Respiratory failure on admission	59 (55%)
Ventilatory support during ICU admission:	
- Invasive ventilation	74 (69%)
- Non invasive ventilation	19 (10%)
Mechanical ventilation duration (d, IQR)	1 [1,2,3,4,5,6,7]

**Table 2 jcm-11-07126-t002:** Investigated features and comparisons between SRs and FTRs.

	Tot %	SRs, *n* (%)	FTRs, *n* (%)	*p*-Value *
**Normal appearance of one or more zones**	34 (2%)	28 (47%)	6 (5%)	<0.0001
**A-lines**	63 (37%)	39 (66%)	24 (21%)	<0.0001
**B-lines**	96 (56%)	38 (64%)	58 (52%)	0.1446
No additional description	28/96 (29%)	17/38 (45%)	11/58 (19%)	0.0109
Number of B lines/B lines	31/96 (32%)	13/38 (34%)	18/58 (31%)	0.8246
Semiquantitative description few, many, some/B lines	37/96 (39%)	10/38 (26%)	27/58 (47%)	0.0555
Defined, crowding/B lines	16/69 (17%)	5/38 (13%)	11/58 (19%)	0.5796
**Consolidation tissue-like**	56 (33%)	26 (44%)	30 (27%)	0.0264
Consolidation small, large, etc.	8/56 (14%)	5/26 (19%)	3/30 (10%)	0.4507
Consolidation measure in cm	2/56 (4%)	1/26 (4%)	1/30 (3%)	1.0000
**Subpleural consolidation**	13 (8%)	6 (10%)	7 (6%)	0.3750
**Bronchogram**	20 (12%)	12 (20%)	8 (7%)	0.0218
**Bronchogram/consolidations**	20/56 (36%)	12/26 (46%)	8/30 (27%)	0.1666
**Atelectasis**	40 (23%)	19 (32%)	21 (19%)	0.0582
Defined as minimal, initial, small, large, extended, etc.	9/40 (23%)	4/19 (21%)	5/21 (24%)	1.0000
**Pneumonia**	12 (7%)	11 (19%)	1 (1%)	<0.0001
Quantification	NA	NA	NA	
**Pneumothorax**	2 (1%)	2 (3%)	0 (0)	0.1177
**Effusion**	69 (40%)	22 (37%)	47 (42%)	0.6239
Presence without any other specification	3/69 (4%)	3/22 (14%)	0/47 (0)	0.0294
Small, minimum, large, abundant	44/69 (64%)	10/22 (45%)	34/47 (72%)	0.0363
Measures (depth in cm or caudal extent; i.e., *n° of intercostal spaces*)	31/69 (42%)	11/22 (50%)	20/47 (43%)	0.6106
Estimation of volume	14/69 (20%)	13/22 (59%)	1/47 (2%)	<0.0001
Appearance/echogenicity	5/69 (7%)	0/22	5/47 (11%)	0.1694
**Curtain sign**	9 (5%)	2 (3%)	7 (6%)	0.7203

Values are expressed as *n* (%); * Fisher exact test.

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
