# Peer review of "Thoracic UltrasONOgraphy Reporting: The TUONO Study"

_jcm, 2022, doi:10.3390/jcm11237126_

Round 1
Reviewer 1 Report
This paper is related to lung ultrasound reporting in Italian ICUs. The paper was well-designed and written but the main limitation is low number of reporting. During nearly two years and from 21 ICUs where lung ultrasound has been routinely used in daily care, there were only 171 reports. It wasn't also stated that how many ultasound examinations were performed during study period or the rate of reports was low or high? With this number, it would be quite uncertain whether the reporting can change clinical practice or identify potential benefits.
Author Response
This paper is related to lung ultrasound reporting in Italian ICUs. The paper was well-designed and written but the main limitation is low number of reporting. During nearly two years and from 21 ICUs where lung ultrasound has been routinely used in daily care, there were only 171 reports.
We are thankful to the reviewer for his suggestions, which helped us improve and present our manuscript more effectively.
Regarding the first point, even though the collection period was extended to 18 months, data were collected from 21 ICU during only 7 days per ICU, as some ICUs were able to perform data collection earlier and others joined the study later. Thus, 171 exams are to be considered over 7 days over 21 ICUs.
We now further clarify this statement in the following paragraph in the methods section:
“We collected all the LUS reports performed in the enrolled ICUs during 7 days (“index week”) selected by each center in the timespan between January 25th, 2018 and November 17th, 2019. Data were collected and reported by the center on a predefined form in a dedicated section of Prosafe software (Prosafe Core, GiVITI, Ranica - Bergamo, Italy), a software created by GiVITI for data collection for quality improvement and scientific research in intensive care. For this study, a new Prosafe software interface stem was built to allow centralized reports collection while preserving the exact structure of reports.
During the index week, all the participating centers were required to send all the LUS exams performed and documented in clinical record, even if LUS reporting was partial or incomplete”.
It wasn't also stated that how many ultasound examinations were performed during study period or the rate of reports was low or high? With this number, it would be quite uncertain whether the reporting can change clinical practice or identify potential benefits.
Thank you for this comment. Over the 7 days of study, each ICU contributed with an average number of 8 exams, with an interquartile range of one to nineteen, about 1 to 3 LUS exam per day.
All the ultrasound examinations performed in these ICU during these 7 days were reported in the study (100% of reporting). As the purpose of this study was to evaluate real world documentation of lung ultrasound examinations in clinical records, we did not ask physicians to increase the number of LUS during the study week; rather, they were instructed to maintain their clinical practice and reporting standards.
As correctly identified by the expert reviewer, poor reporting is indeed one of the limitations in the diffusion of daily adoption of LUS, and is also limiting clinical practice, and the object of this study is to expose the current limitations in real world adoption of LUS and promote further studies to overcome them.
Reviewer 2 Report
Dear colleague
Thank you for the opportunity to revew this manuscript in which the author and team reviewed the LUS reports performed across 21 Italian ICUs. In particular, comparing the structured vs free-text formats of report.
There are some minor grammatical errors but overall, the manuscript is clear and readable.
My key comments/considerations are around the methodology of the study.
- The authors mentioend that the data were collected on a predefined form using a dedicated software. Does this mean that the health records from these various institutions are on the same platform or merely the search was conducted using the same platform? Further expanding on this would improve the generalisability of the manuscript as different ICUs (especially outside Italy) would be able to understand the landscape.
- How did the authors ensure that all LUS reports were captured? Were they all recorded under a separate 'ultrasound/imaging section'? In some institutions, LUS are performed as part of the clinical examination and hence its finding are incorporated into this rather than a separate heading?
Other comments
- what is the overall intention of the study - encourage standardisation of reporting?
- 1 of the main strengths of LUS is that it is focussed, with a particular clinical question to be answered. Hence, especially in the case of the FTRs, the reason why certain terms were not included in the report is that it wasn't the clinical question. Eg. if the ultrasound was performed to confirm the size of the effusion, it is perhaps unsurprising that the term 'diaphragm' did not appear. It would be perhaps unfair to label these as 'incomplete'.
- Did the authors get a sense of how many of these scans were done 'routinely' as oppose to a specific clinical question?
- As a follow up to the above question, how many of the scans resulted in change of management or unexpected findings?
- The authors mentioned that the study maybe disproportionate for centres that perform a lot of LUS, could the authors perhaps give an idea/results of the number of LUS scans performed by each contributing ICU?
Author Response
Thank you for the opportunity to revew this manuscript in which the author and team reviewed the LUS reports performed across 21 Italian ICUs. In particular, comparing the structured vs free-text formats of report. There are some minor grammatical errors but overall, the manuscript is clear and readable.
We are grateful to the reviewer for this acknowledgment, and for the comments which allowed us to improve the manuscript. We also corrected those minor grammatical errors.
My key comments/considerations are around the methodology of the study.
The authors mentioned that the data were collected on a predefined form using a dedicated software. Does this mean that the health records from these various institutions are on the same platform or merely the search was conducted using the same platform? Further expanding on this would improve the generalisability of the manuscript as different ICUs (especially outside Italy) would be able to understand the landscape.
Thank you for the comment. We now better explain this in the methods section, as we write:
“Data were collected and reported by the center on a predefined form in a dedicated section of Prosafe software (Prosafe Core, GiVITI, Ranica - Bergamo, Italy), a software created by GiVITI for data collection for quality improvement and scientific research in intensive care. For this study, a new Prosafe software interface stem was built to allow centralized reports collection while preserving the exact structure of reports.
During the index week, all the participating centers were required to send all the LUS exams performed and documented in the clinical records, even if LUS reporting was partial or incomplete.”
How did the authors ensure that all LUS reports were captured? Were they all recorded under a separate 'ultrasound/imaging section'? In some institutions, LUS are performed as part of the clinical examination and hence its finding are incorporated into this rather than a separate heading?
All the participating centers were instructed to select one week over the course of several months (the “index week”); by protocol, they were required to submit all the LUS reports, regardless of their incomplete reporting. The selection of a single index week was a deliberate strategy to increase the adherence of the centers in order to obtain all the examinations performed.
We now further explain this in methods
During the index week, all the participating centers were required to send all the LUS exams performed and documented in clinical record, even if LUS reporting was partial or incomplete.
Other comments:
- what is the overall intention of the study - encourage standardisation of reporting?
Thank you for this comment. The overall intention of the study was to identify characteristics and limitations of current LUS reporting. Identification of the real-world limitations in LUS reporting may be the base for improved and standardized reporting. After acquiring the results of this study, we conducted a consensus conference involving ICU and LUS experts to provide a possible model of a standardized report.
- one of the main strengths of LUS is that it is focused, with a particular clinical question to be answered. Hence, especially in the case of the FTRs, the reason why certain terms were not included in the report is that it wasn't the clinical question. Eg. if the ultrasound was performed to confirm the size of the effusion, it is perhaps unsurprising that the term 'diaphragm' did not appear. It would be perhaps unfair to label these as 'incomplete'.
We agree with the reviewer with this suggestion. We did not mean to suggest that physicians should look and record all possibile findings every time they perform LUS, but we advise that a detailed description is pivotal for subsequent comparisons in the evaluation of the evolution of a pathological condition affecting the critical patient.
Therefore, a complete examination has a different value compared to a focused one: the latter is by definition aimed at looking for one or few elements and remains a cornerstone in daily practice, especially in the emergency setting.
- did the authors get a sense of how many of these scans were done 'routinely' as oppose to a specific clinical question?
Thank you for this question. As we express in the results section, 21 (11%) reports were performed to rule out specific conditions. We also specify the percentage of examinations by reason for LUS assessment in the supplementary material - table A.
- as a follow up to the above question, how many of the scans resulted in change of management or unexpected findings?
We appreciate this comment. We now better specify this aspect in the results sections:
“LUS results guided the implementation of new therapeutic strategies, mostly involving ventilation, CRRT, antimicrobial and diuretic therapy, namely in 19% of SRs and 6% of FTRs (p =0.01)”.
- the authors mentioned that the study maybe disproportionate for centres that perform a lot of LUS, could the authors perhaps give an idea/results of the number of LUS scans performed by each contributing ICU?
Thank you for this suggestion. We now report in the results section the number of US performed by each ICU per week, expressed as a mean and an interquartile range, as we write “The number of LUS performed by each ICU during the index week, expressed as a mean and an interquartile range, was 8 [1-19]”.
Round 2
Reviewer 1 Report
In addition to my previous review, I am still not sure how this paper will contribute to the literature with this number of lung reports as many meta analysis were also published regarding the lung ultrasound.